# Working in labor and delivery unit increases the odds of work place violence in Amhara region referral hospitals: Cross-sectional study

**Eyaya Habtie Dagnaw** [1] *, **Abrham Walelign Bayabil**[1], **Tigist seid Yimer**[1], **Tewodros Seyoum Nigussie**[2]

**1** Department of Midwifery, College of Medicine and Health Sciences, Debre Tabore University, Debre Tabore, Ethiopia, **2** Department of Midwifery, College of Medicine and Health Sciences, University of Gondar, Gondar, Ethiopia

* eyuhabt143@gmail.com

**Data Availability Statement:** All relevant data are within the manuscript and its Supporting Information files.

## Abstract

### Background

Workplace violence is any act of negative behavior that causes, physically and psychologically harm to health professionals face in the workplace. The prevalence of workplace violence becomes a challenging occupational issue with increasing nature worldwide. In spite of the seriousness and the impact of the problem, little is known about its magnitude and determinants in the study area and even in Ethiopia.

### Objective

The study aimed to assess the magnitude of workplace violence and its associated factors among health care providers working for the last one year at Obstetrics and gynecology department in Amhara Regional State Referral Hospitals, Ethiopia 2019.

### Methods

Institutional based cross-sectional study was conducted from October 1st to 30th, 2019. 503 study participants were incorporated in the study. A pre-tested structured questionnaire was used to collect the data. Data were entered into EPI info version 7.2.3.1 and analyzed using SPSS version 23. Binary Logistic regression model was fitted to identify factors associated with workplace violence considering the association to be significant p- value <0.05.

### Result

This study revealed that 44.5%of the health care providers had reported workplace violence (95% CI: 40.2–48.7). Of this majority of the Victims were experienced a verbal type of violence 200 (88.1%), followed by physical 14 (6.2%), sexual 11 (4.8%), and racial two (0.8%). Factors of workplace violence in this research with statically significant, were: working in labor ward (AOR = 7.4,95% CI: 2.9–18.7), Female sex of participant (AOR = 2.4, 95%

**Funding:** The research was supported in part by University of Gondar, College of Medicine and Health Science for data collection not for publication cost. This agency did not have a role in the design; collection, analysis, and interpretation of data; or in writing the manuscript.

**Competing interests:** No authors have competing interests.

**Abbreviations:** ICN, International Council of Nurses; ILO, International Laborers Organization; MSc, Master of Science; OSHA, Occupational Safety and Health Association; OBGYN, Obstetrics and Gynecology; PSI, Public Service Institution; SPSS, Statistical Product of Science Solution; WHO, World Health Organization; WPV, Workplace Violence.

CI:1.4–4), work experience less than 5 years(AOR 8.5, 95%CI:7.3–33.3) and numbers of staff less than5 in a shift (AOR = 5.3 95% CI:3.8–39.8) and 5–10 staff in a shift (AOR = 3.3, 95% CI:2.7–25).

## Conclusion and recommendations

The prevalence of workplace violence among obstetrics and gynecology department health professionals in Amhara regional state referral hospitals was high. Developing an incident resolution protocol and legislations to encourage health professionals to prompt report violent acts and judicial punishment of perpetrators will be useful to combat workplace violence at obstetrics and gynecology department.

## 1. Introduction

Workplace violence (WPV) is defined as any form of negative behavior that causes physical and psychological harm to health professionals at workplace. It includes direct physical assaults, written or verbal threats, physical or verbal harassment, which involves an explicit or implicit obstacle to the safety, well-being of health care workers. In short, term words and actions that hurts health care providers at work [1, 2]. As works of literature showed, workplace violence in health care settings becomes a major problem and challenging occupational issue with increasing nature. The source of violence can be from clients and attendants, senior physicians, companions, immediate bosses and other administrative staff [3, 4].

According to International Laborers Origination (ILO), among all other workplaces, 25% of violence accidents occur in the health care institutions; almost half of the health workers have experience of violence, and many of them are suffering from physical and psychological health morbidities. Workplace violence-related injury is the second leading cause of occupational morbidity and the third leading cause of traumatic injuries that accounts for 16% of more than 6.5 million acts of violence experienced by individuals [4–6]. The issue has become an emerging problem in different health sector settings and among health care providers of both industrialized and developing countries. Workplace violence is being more serious for front line caregivers including nurses and midwives also taken as part of the job. Even though, there is no a summation national report in Ethiopia, workplace violence is ranging from 26.7 to 58.2% based on a reports of individual articles [7–11].

The impact of violence in the workplace is extends to all component of health care system causing various problems like stress, physical morbidity, burn out; poor job satisfaction, high turnover of professionals, leads negligent of health care providers, ineffective clinical performance cumulatively result in compromised quality health care and the ultimate victims are the patients [11–13].

Factors that related to workplace violence more or less listed by some scholars which can be categorized into three classifications; the first one is environmental or working unit problems such as long waiting time, frequent interruptions uncertainty regarding patient care and treatment, heavy workload, numbers of staff in the shift, and working in night shifts. The second one is related to organizational factors like inefficient teamwork among staff, organizational injustice, and lack of aggression management programs, incident reporting procedures and violence dealing process. The third category of violence includes factors related to personal and psychosocial issues like being female, younger age, being inexperienced less than 5 years, previous history of violence and other sociodemographic factors [13–16]. Even though

different kinds of literature showed an increased prevalence of workplace violence its magnitude and associated factors of Workplace violence, in Obstetrics and gynecology department care providers were not assessed as such in the study area even in Ethiopia.

Therefore, this study aimed to assess the magnitude and associated factors of workplace violence among health care providers working at obstetrics and gynecology care units in Amhara Regional State Referral Hospitals.

## 2. Methods

A cross-sectional study was conducted in Amhara regional state referral hospitals from October 1st to 30th, 2019. Amhara regional state is found in the northern part of Ethiopia. There are six public referral hospitals in the region. These are University of Gondar Referral Hospital, Felege Hiwot Referral hospital, Tibebe Gion Referral Hospitals, Debre Markos Referral Hospital, DebreBirhan Referral Hospital, and Dessie referral hospital. These hospitals each serving around 8 million people in their catchment area. Around 520 health professionals working in the OBGYN department from various professions midwifery, nurses, integrated surgical officers, General practitioners, OBGYN residents and senior specialties of OBGYN.

### 2.1 Study population

All health care providers working in the obstetrics and gynecology department in Amhara Regional State Referral Hospitals during the data collection period. Inclusion criteria All health professionals working in OBGYN department in Amhara Regional State referral hospitals.

### 2.2 Exclusion criteria

OBGYN Residents other than referral hospitals, intern and other students and health care staff less than six-month stay in the hospitals were excluded.

The actual number of providers working in OBGYN department in Amhara region referral hospitals was around 520 providers. So all health professionals were included in the study.

### 2.3 Variables of the study

Workplace violence (YES or NO) is the dependent variable. Independent variables Psychosocial and sociodemographic factors: (Age, sex, religion, profession, ethnicity, Year of experience, monthly income, educational level, frequency of violence, self-rated communication skill, self-rated clinical skill, job satisfaction, and time of recent violence). Working unit factors: (Number of staff members, Working unit, Night shift, Types of profession Numbers of a health profession in a shift). Organizational factors: (Violence dealing measures, Incident reporting procedure, and Presence of specific policy for WPV, facility type, Human resource)

### 2.4 Data collection procedure

Data were collected by self-administered structured and pretested questionnaire. The questionnaire is adapted from the International Labor Office (ILO) International Council of Nurses (ICN) World Health Organization (WHO), and Public Services International (PSI) joint program survey questionnaire [17]. One professional midwife for each hospital was assigned to collect the data and two MSc midwives were assigned for supervision a total of eight persons for all hospitals. The questionnaire was developed in English and translated to Amharic version. Then back to English by language experts for the sake of consistency. The questionnaire was prepared from internationally recognized organizations (WHO, ILO, ICN, and PSI) and literature then translated into Amharic then back to English with a help of

language experts to ensure consistency and in the phrasing of questions to maximize accuracy. Training was provided for data collectors and supervisors for one day about the purpose of the study and techniques of data collection. The trained data collectors were supervised during data collection, and each questionnaire was check for completeness daily. Two individuals at two different computers to minimize error did the data entry and the similarity of data entered was checked. The questionnaire was pre-tested to check the response, language clarity and appropriateness of the questionnaire while the pretest was done at Debetabor hospitals with 5% of sample size. At the end of the pretest, vague words and incoherence of the tool were identified and correction measure was taken.

After the data has checked for completeness and accuracy, it was coded manually and then entered into Epi Info version 7.2.3.1 and exported to SPSS version 23 for analysis. Descriptive statistics were performed on numerical value, mean, frequencies, proportion to describe the study population about dependent and independent variables. The binary logistic regression model was fitted to determine the crude associations of each independent variable with workplace violence. Those variables with a p-value of less than 0.2 were entered for multivariable logistic regression for further analysis to control for confounding variables. Adjusted OR with 95% CI was used to estimate the strength of the association between independent variables associated with the outcome variable. Finally, variables with p-value <0.05 were considered significantly associated with workplace violence. The Hosmer and Lemeshow goodness of fit test was used to check model fitness and indicates that the data is fit for the model (p = 0.63).

## 2.5 Ethics approval and consent to participants

Ethical clearance was obtained from Ethical Review Committee of the school of midwifery on the behalf of the Institutional Review Board (IRB) of the University of Gondar. A letter of cooperation was written from UOG to Amhhara Regional Health Bearu and each referral hospitals.

The objective and the aim of the research was explained to the study subjects on the information sheet of the questionnaire, written information as a consent was attached from consent form that explains the purpose and overall information of the study. Participants of the study were signed for their involvement (to be a participant) was after their complete consent. Any health professionals working in the OBGYN department who were not willing to participate in the study were not forced to participate, no personal identifications were included in the data sheet and all data taken from the participants were kept strictly confidential and used only for the study purpose.

## 3. Result

### 3.1 Participants' socio demographic and psychosocial characteristics

Out of the expected 520 participants, 503 gave a complete response with a response rate of 96.7%. The mean age of the participants was 28.8 (SD ± 3 .3) years. More than three fourth 84. % (423) of the study, participants were between the age of 25–35 years. About 43.5% (219) of the study participants were single. In the religion aspect nearly, two-third of the respondents were Orthodox followers 65% (327) followed by Muslim 23.5% (118). Around three–fourth 74.8% (376) of the participants were Midwives followed by nurses (12.9%) (Table 1).

### 3.2. Organizational and working unit information

Among 503 participants, 85% (428) of them did not know whether there was an institutional policy or not in their institution. More than half of 56.5% (284) of participants did not know the presence of WPV dealing team (Table 2).

**Table 1. Psychosocial and demographic characteristics among OBGYN department health professional staff of Amhara regional state referral hospital Ethiopia 2019 (n = 503).**

| Variables | | Frequency | Percentage (%) |
|---|---|---|---|
| **Age in years** | | | |
| <25 | | 69 | 13.7 |
| 25–35 | | 423 | 84.1 |
| >35 | | 11 | 2.2 |
| **Gender** | | | |
| Male | | 312 | 62 |
| Female | | 191 | 38 |
| **Religion** | | | |
| Orthodox | | 327 | 65 |
| Muslim | | 118 | 23.5 |
| Protestant | | 45 | 8.9 |
| Others* | | 13 | 2.6 |
| **Marital status** | | | |
| Single | | 219 | 43.5 |
| Married | | 270 | 53.7 |
| Others** | | 14 | 2.8 |
| **Educational status** | | | |
| Diploma | | 16 | 3.2 |
| Degree | | 430 | 85.5 |
| M.Sc. | | 26 | 5.1 |
| OBGYN Specialist | | 31 | 6.2 |
| **Ethnicity** | | | |
| Amhara | | 465 | 92.4 |
| Tigray | | 14 | 2.8 |
| Oromo | | 15 | 3 |
| Others*** | | 9 | 1.8 |
| **Profession** | | | |
| Nurse | | 65 | 12.9 |
| Midwife | | 376 | 74.8 |
| Physician*** | | 57 | 11.3 |
| IESO | | 5 | 1 |
| **Work experience** | | | |
| 1–5 years | | 246 | 48.9 |
| 6 to 10 years | | 144 | 28.6 |
| 11 to 15 years | | 80 | 15.9 |
| >15 years | | 33 | 6.6 |
| Monthly income in USD($) | | | |
| < 100 | | 31 | 6.2 |
| 101–151 | | 228 | 45.3 |
| 1 52–215 | | 150 | 29.8 |
| >215 | | 94 | 18.7 |
| Self-rated communication | Very poor | 56 | 11.1 |
| | Poor | 193 | 38.4 |
| | Good | 25 | 5 |
| | Very good | 229 | 45.5 |

(*Continued*)

**Table 1.** (Continued)

| Variables | | Frequency | Percentage (%) |
|---|---|---|---|
| Self-rated clinical skill | Very poor | 29 | 5.8 |
| | Poor | 161 | 32 |
| | Good | 60 | 11.9 |
| | Very good | 253 | 50.3 |
| Job satisfaction | Very dissatisfied | 201 | 40 |
| | Dissatisfied | 23 | 4.6 |
| | Satisfied | 40 | 8 |
| | Very satisfied | 239 | 47.5 |

Others ** = Separated, widowed and divorced

Physician = GP, OBGYN residents, and OBGYN specialist

Others*** = kimant, agew, hadiya, welyta, gamgofa

Diploma = a trained health profession minimum of 3 years after completion of secondary school

### 3.3 Prevalence of workplace violence among OBGN health professionals

The overall workplace violence in this study was found to be 224 (44.5%) (95% CI: 40.2–48.7). Of this majority of the victims, 88.1% (200) were experienced a verbal type of violence (Fig 1).

From those victims of verbal violence, threat(verbal/written) was 25.7%(52), repeated disturbance 15.3%(31), mobbing/bullying 12.9%(26), any level of disrespect 22.8%(46) and abuse 23.3%(47) were observed. From those reported physical violence subtypes half of them were complaining of physical injury 50% (12).

**Table 2. Organizational and working unit information among Amhara regional state referral hospitals OBGYN department health professional staff, Ethiopia, 2019.**

| Variables | Frequency | Percent | |
|---|---|---|---|
| Number of care providers in a shift | <5 | 132 | 26.2 |
| | 5–10 | 223 | 44.3 |
| | 11–15 | 105 | 20.9 |
| | >15 | 43 | 8.5 |
| Working in shifts | Yes | 270 | 53.7 |
| | No | 233 | 46.3 |
| Presence of policy for WPV | Yes | 5 | 1 |
| | I don't know | 428 | 85.1 |
| | No | 70 | 13.9 |
| Working from 6 pm to 7 am | Yes | 47 | 15.1 |
| (In night shift) | No | 252 | 84.3 |
| Presence WPV Incident reporting form | | | |
| | Yes | 22 | 4.4 |
| | No | 197 | 39.2 |
| | I don't know | 284 | 56.5 |
| Presence WPV dealing team | Yes | 10 | 2 |
| | No | 184 | 36.6 |
| | I don't know | 309 | 61.4 |

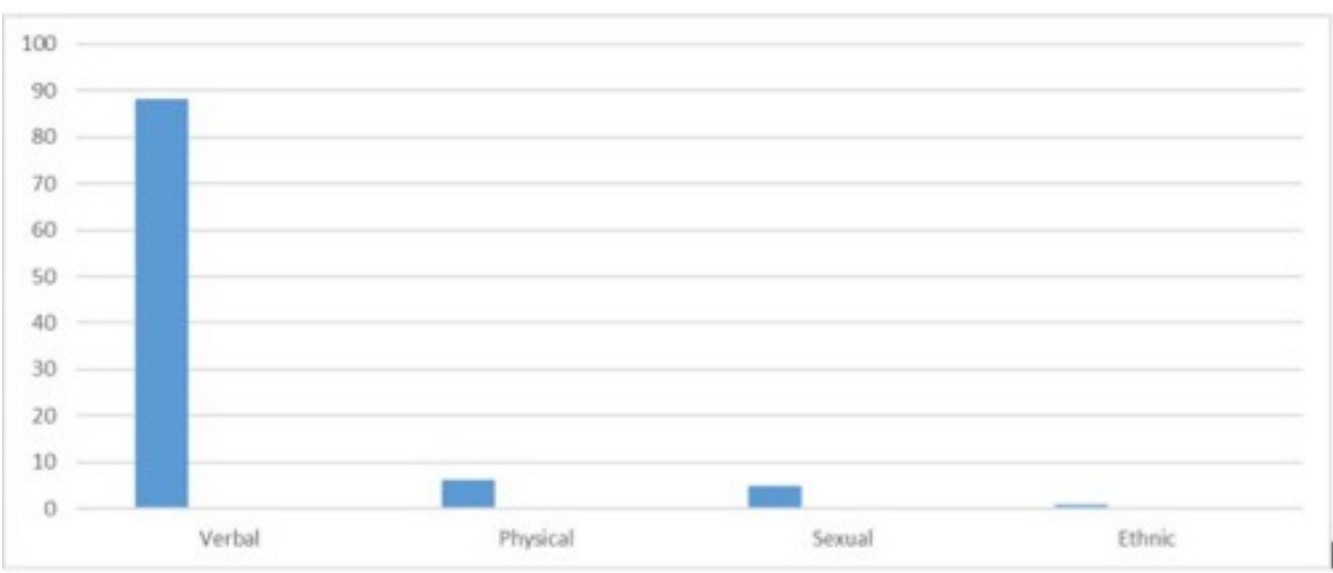

**Fig 1. Forms of workplace violence among Amara regional state referral hospitals OBGYN department health professional staff, Ethiopia 2019.**

### 3.4. Perceived reasons for workplace violence

The perceived reason for work place violence mentioned in this study were listed as follow, I didn't know 47.1%(107), followed by dissatisfaction from patient care and treatment 30.4% (69), miscommunication 11.9%(27), and long waiting time 6.6%(15).

Regarding the place where they have been attacked, half of the victims were attacked in wards 51.8% (116) followed by staff room, 24.6% (55), the rest were somewhere in the compound 19.6% (44), and at Outpatient department 4% (9). The majority of victims were experienced WPV before 6 months 53.1% (119) in last 12 months and 105(46.9) of them reported violence as it happened within the last 6 months. Among victims, 40.6% (91) of them reported as they face WPV sometimes followed by once 80(35.7%) and at all time23.7percentage (53). The reactions of victims when 9% 44% 41 6%.

Subjects of attackers in this research patient/client Colleague Attendants Administrative/ Management in decreasing order (Fig 2).

In 96.9% (217) occasions nothing happened to the attackers, the rest 1.8% (4) took administrative action, 0.9% (2) verbal warning was issued, and 1(0.4%) reported to senior. 78.1% (171) participants who had WPV thought that the incident could have been prevented, and 23.7% (53) thought may not be prevented. Of all victims, only 4% (9) given support and majority 96% (215) had not given any support. According to different characteristics of respondents, the prevalence of workplace violence was assessed and it shows that among all professions, Midwifery profession was 80.4% (180) and regarding the working unit prevalence of WPV who are working in labor and delivery was 33% (166).

### 3.5. Factors associated with workplace violence among OBGYN department staff

In bi-variate logistic regression sex, profession, educational status, monthly income, working unit, working in night shift, numbers of staffs in a shift, work experience, and participants age were significantly associated; however, in multiple logistic regression working unit, sex of participant, work experience and numbers of staffs in a shift remain significantly associated with workplace violence. Participants working in labor and delivery unit (AOR = 7.4, 95%

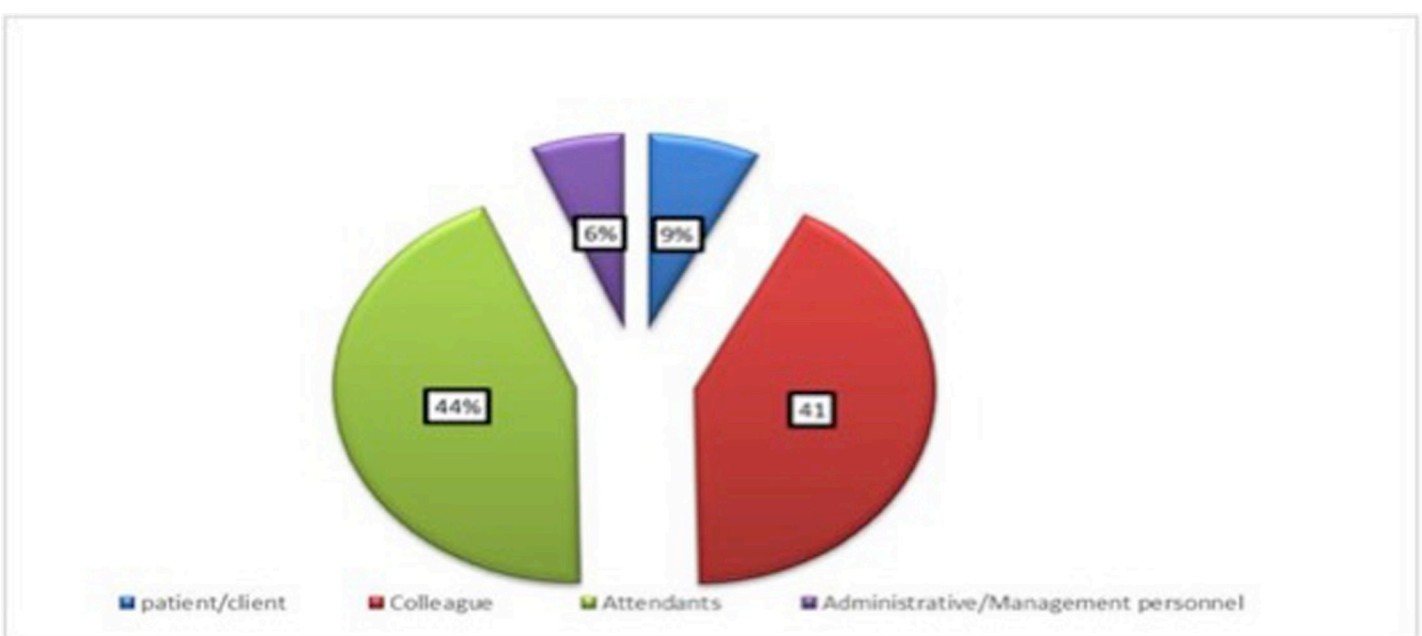

**Fig 2. Showing attackers among Amhara regional state referral hospitals OBGYN department health professional staff, Ethiopia, 2019.**

CI: 2.9–18.7), being female (AOR = 2.4, 95% CI 1.4–4), work experience in years (< 5 years AOR = 8.4,95%CI:1.3–33.3,) and numbers of staffs in a shift (< 5 AOR = 5.3, 95%CI: 3.8–39.8 and 5–10; AOR = 3.3, 95% CI: 2.7–25) were independently with associated to workplace violence (Table 3)

## 4. Discussion

Currently, health professionals over the world are grumbling of their profession due to increasing features of violence at their working place [18].

Workplace violence against health professionals in this research is categorized as a type II and type III (perpetrator is a customer or patient of the workplace and current worker of the institution) [11].

The finding of this research showed that the prevalence of workplace violence was about 44.5% [95% CI: 40.2–48.7]. This finding was high compared to a report by Occupational safety, health, and administration (OSHA) 2016 [13]. The possible reason might be as it was incorporated both developed nations with sophisticated strategies and policies, and the report was a general health care providers but this result was specific to the obstetrics and gynecology unit. This study also higher than a study conducted in China among nurses working at obstetrics ward and it was about 38.3% [13]. This difference could be due to the sociodemographic, cultural difference, individual and organizational factors among the perpetrators and country setting. Again, this study was higher than domestic studies that were done in Amhara regional state referral hospitals workplace violence among nurses 4 years ago and a study done at Hawasa city administration health facilities 26.7% and 29.9% [10, 11]. The difference might be due to the study population and study setting. On the contrary, the current finding was less than a study conducted in Korean shows the magnitude of violence was found to be 88.3%, Saudi Arabia 57.5%, Israeli 58% and in Turkey indicate, that 81.8% [19–22]. The possible reason might be the socio-cultural difference among perpetrators study participants. This study

**Table 3. Bi-variate and multivariable logistic regression analysis of factors associated with workplace violence among health care providers working OBGYN department in Amhara regional state referral hospital, Ethiopia 2019.**

| Variables | | WPV | | Odds ratio with 95% CI | |
|---|---|---|---|---|---|
| | | Yes | No | COR | AOR |
| Gender | Male | 107 | 205 | 1 | 1 |
| | Female | 117 | 74 | **3 (2.4–4)** | **2.4 (1.4–4)**** |
| Age | < 25 years | 37 | 32 | 1.52 (0.2–28) | |
| | 25–35 years | 187 | 247 | 1 | |
| Working unit | ANC Outpatient department | 15 | 52 | 1 | 1 |
| | FP & EPI | 8 | 56 | 0.49(0.19–1.26) | 0.7 (0.25–1.98) |
| | GYN Outpatient department | 8 | 50 | 0.55(0.21–1.42) | 0.76 (0.23–2.1) |
| | GYN inpatient department | 27 | 64 | 1.46 (0.7–3) | 1.6 (0.7–3.7) |
| | Labor and Delivery | 166 | 57 | **10 (5–19.3)** | **7.4 (2.9–18.7)**** |
| Profession | Nurse | 22 | 43 | 0.93 (0.4–1.9) | |
| | Midwife | 180 | 196 | 1.6(0.9–2.9) | |
| | Physician | 22 | 40 | 1 | |
| Work experience | 1–5 years | 161 | 85 | **28.7 (6.8–125)** | **8.4 (1.3–33.3)*** |
| | 6 to 10 years | 56 | 88 | 10(2.3–42.8) | 7.2 (0.6–16.8) |
| | 11 to 15 years | 7 | 106 | 1 | 1 |
| Educational level | Diploma | 11 | 5 | 4 (1.1–14) | |
| | Degree | 196 | 234 | 1.5 (0.7–3) | |
| | MSc | 6 | 20 | 0.5(0.17–1.7) | |
| | Specialty | 11 | 20 | 1 | |
| No of staff in a shift | < 5 | 78 | 54 | **6.3 (2.7–14.6)** | **5.3 (3.8–39.8)**** |
| | 5–10 | 116 | 107 | **4.7 (2.1–10)** | **3.3 (2.7–25)**** |
| | 11–15 | 22 | 83 | 1.2 (0.47–2.8) | 1.48 (0.4–4.7) |
| | >15 | 8 | 35 | 1 | 1 |
| Monthly income | <100 USD | 18 | 13 | 4.5(1.9–10.6) | |
| | 101–151 USD | 137 | 91 | 4.9(2.8–8.5) | |
| | 152–215 USD | 47 | 103 | 1.4(0.8–2.6) | |
| | >215 USD | 22 | 72 | 1 | |
| Working in night shift | Yes | 156 | 159 | 1.7 (4.8–10.8) | |
| | No | 68 | 120 | 1 | |

also less than a finding of WPV among nurses in Gambia hospitals, which was 62.1% [7]. The possible difference might be due to study subjects that it was among nurses working at the maternity ward.

This finding was less than local researches done in Gondar administration city workplace violence among health care providers working private and governmental health institutions and a study done in Oromia regional state referral hospitals workplace violence among health care providers which were 58.2% and 82.8% respectively [8, 23]. The difference might be due to study population they used both private and governmental institutions as well as they incorporated all health care worker in their study working in all health care units unlike the current study. A research done in Amanuel Mental Specialized Hospital in Addis Ababa, workplace violence is greater than this study. As it shows, verbal violence was 62.1% physical violence 36.8%, and 28 sexual violence 21.8% [9]. The possible reason might be due to nature of health condition of clients they are caring for. Among all other professions in the department prevalence of WPV 80.4% accounts for midwifery profession. This finding is consistent with study done in Turkey [24]. The possible reason might be that midwives are the front line caregivers

in the department. This study identified that factors associated with workplace violence were the sex of participant (female), working unit (labor and delivery suite), and work experience (< 5 and 5–10). In this research, the odds of workplace violence were 2.4 times more likely in female health care providers. This finding is in line with a study done at Hawasa city administration health facilities and a study conducted in Egypt Cairo hospitals [6, 10, 25]. The reason might be due to traditional perceptions of taking male as superior beyond female. However; it is different from a study conducted among nurses in Amhara regional state referral hospital revealed that male nurse are more victim of physical violence compared to their counter female colleagues [26]. The difference could be attacking females taken as breaking social norm, the working department and the study subjects also taken as the possible reason as it was only conducted among nursing staff.

Work experience of the health care providers was found to be a risk factor for workplace violence in different literature in this research also it was found that the odds of workplace violence 8.5 times more likely higher for whom health care providers work experience was 1 to 5 years [8, 11, 12]. This finding was in line with the study done at Sidama zone southern Ethiopia [10]. The reason might be methodological similarity and possible reason for the factor might be, less experienced and less exposed to clinical care are less effective in clinical and communication skills. Workplace violence magnitude is not the same across different working units and wards as we revised different articles even though there is no study specific to the labor and delivery unit as far as my knowledge. In this research, the odds of workplace violence are 7.4 higher in health care providers working at the labor and delivery unit. The reason could be due to the labor and delivery unit is more stress full and anxious area for the mothers, to the attendants and as well to the care providers. This is known that numbers of care providers allocated to each working unit within a shift based on the workload of the unit. Kinds of literature found less than 5 and 10 numbers of care providers in a shift have a significant association to have workplace violence; meanwhile, this research found that the odds of workplace violence among numbers of professionals less than 5 were 5.3 times higher as compared to greater than 15 care providers. It was also true that the odds of workplace violence among care providers between five to ten were 3.3 times higher than that of greater than 15 personals in a shift. The reason could be due to dissatisfaction from patient care and treatment due overload of a patient to professional imbalance and long waiting time to a patient receive care. This finding is in line with researches done in Amhara regional state referral hospitals [11].

## 5. Limitation of the study

In such types of research recall bias may encounter since respondents may not remember whether they had violence or not and how many times they encounter violence at work. To minimize this challenge the study deals with a one-year duration prevalence.

## 6. Conclusion

After all, the prevalence of workplace violence among obstetrics and gynecology department health care providers in Amhara regional state referral hospitals was high as the finding of research showed. Sex of the participant (being female), working unit (labor and delivery suite), number of staff in a shift (less than 10) and work experience (less than 5 years) of the respondent ware statically associated with the outcome variable independently.

This research revealed that the prevalence of workplace violence was high. Government stakeholders and hospitals may give attention to reduce the burden of the problem for better health care in obstetrics, gynecology, and neonatal health. This can be address by drafting

WPV incident reporting procedures, assign security personnel, making legislations and polices to make the health care place safe and conducive places both for patient and clients

## 7. Recommendation

### To the ministry of health and Amhara regional state health bureau

Drafting of incident resolution protocol may help in encouraging health professionals to promptly report violent acts.

Developing legislations for judicial punishment of perpetrators may also helpful.

### To the hospitals

The presence of incident reporting procedures at the institutions may help to reduce the problem.

The establishment of violence dealing team from various professions may be helpful to reduce the incidence.

Give training and creating awareness to the professionals about workplace violence.

Allocate reasonable and adequate numbers of staff in each shift.

Human resource development might help to overcome the incident.

### To Researchers

Further observational approach (mixed study) on violence in the OBGYN department may help to investigate more factors associated with workplace violence.

It is recommended to address the impact of workplace violence at obstetrics and gynecology department.

## Supporting information

**S1 File.**
(DOCX)

**S2 File.**
(DOCX)

## Acknowledgments

We would like to thank University of Gondar for ethical clearance. Our gratitude also great to all data collectors and study participants. We are also glad to thank Amhara Regional Health bureau for writing permission letters to Referral hospitals.

## Author Contributions

**Conceptualization:** Eyaya Habtie Dagnaw, Tewodros Seyoum Nigussie.

**Data curation:** Eyaya Habtie Dagnaw.

**Formal analysis:** Eyaya Habtie Dagnaw, Abrham Walelign Bayabil.

**Methodology:** Eyaya Habtie Dagnaw, Abrham Walelign Bayabil, Tigist seid Yimer, Tewodros Seyoum Nigussie.

**Software:** Tigist seid Yimer, Tewodros Seyoum Nigussie.

**Supervision:** Eyaya Habtie Dagnaw.

**Writing – original draft:** Eyaya Habtie Dagnaw, Tigist seid Yimer.

**Writing – review & editing:** Eyaya Habtie Dagnaw, Abrham Walelign Bayabil, Tigist seid Yimer.

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
