## [Decision Letter · Decision Letter 0]

5 Feb 2021

PONE-D-20-30094

Working in labor and delivery unit increases the odds of work place violence in Amhara region referral hospitals: Cross-sectional study

PLOS ONE

Dear Dr. Dagnaw,

Thank you for submitting your manuscript to PLOS ONE. After careful consideration, we feel that it has merit but does not fully meet PLOS ONE’s publication criteria as it currently stands. Therefore, we invite you to submit a revised version of the manuscript that addresses the points raised during the review process.

The manuscript has been evaluated by two reviewers, and their comments are available below. You will see the reviewers have commented on the interest of your study. However, the reviewers have also raised critical concerns and the manuscript will need significant revision before it can be considered for publication – you should anticipate that the reviewers will be re-invited to assess the revised manuscript, so please ensure that your revision is thorough. I have outlined some of the key concerns noted by the reviewers below, but you should respond to all concerns mentioned by the reviewers in your response-to-reviewers document. 

The key concerns noted by the reviewers relate to updated references and details about the questionnaire. Specifically, Reviewer 2 requested additional information about adaptation, validation, and limitations of the questionnaire. These issues have limitations for the interpretation of the results and should be explored.

Reviewer 1 has recommended that you cite specific previously published works. As always, we recommend that you please review and evaluate the requested works to determine whether they are relevant and should be cited. It is not a requirement to cite these works. 

We look forward to receiving your revised manuscript.

Kind regards,

Danielle Poole

Academic Editor

PLOS ONE

Journal Requirements:

2. Please provide additional details regarding participant consent. In the ethics statement in the Methods and online submission information, please ensure that you have specified whether consent was written or verbal/oral. If consent was verbal/oral, please specify: 1) whether the ethics committee approved the verbal/oral consent procedure, 2) why written consent could not be obtained, and 3) how verbal/oral consent was recorded. If your study included minors, please state whether you obtained consent from parents or guardians in these cases. If the need for consent was waived by the ethics committee, please include this information.

4. Please include additional information regarding the survey or questionnaire used in the study and ensure that you have provided sufficient details that others could replicate the analyses. For instance, if you developed a questionnaire as part of this study and it is not under a copyright more restrictive than CC-BY, please include a copy, in both the original language and English, as Supporting Information.

5. Thank you for stating the following in the Funding Section of your manuscript:

"The research was supported by a grant from University of Gondar, College of Medicine and

Health Science. The granting agency did not have a role in the design; Collection, analysis, and

interpretation of data or; in writing the manuscript."

"NO"

7. Please ensure that you refer to Figure 2 in your text as, if accepted, production will need this reference to link the reader to the figure.

8. Please include a copy of Table 4 which you refer to in your text on page 8.

Reviewers' comments:

Reviewer's Responses to Questions

**Comments to the Author**

1. Is the manuscript technically sound, and do the data support the conclusions?

Reviewer #1: Yes

Reviewer #2: Partly

2. Has the statistical analysis been performed appropriately and rigorously? 

Reviewer #1: Yes

Reviewer #2: Yes

3. Have the authors made all data underlying the findings in their manuscript fully available?

Reviewer #1: No

Reviewer #2: Yes

4. Is the manuscript presented in an intelligible fashion and written in standard English?

Reviewer #1: Yes

Reviewer #2: No

5. Review Comments to the Author

Reviewer #1: Thank you for the opportunity to review this article.

The authors proposed a cross-sectional study to assess the magnitude of workplace violence and its associated factors among health care providers working for the last one year at Obstetrics and gynaecology department in some Ethiopian Regional State Referral Hospitals.

The topic is very current and interesting. I am not a native English speaker, but I find that the article is presented in an intelligible fashion and is written in clear and unambiguous scientific English but, a spelling or syntactic check of the text may be useful.

I find some critical points that must be addressed and resolved before publication.

Here are my suggestions:

#1 Page 9, lies 23-25: The introductory section is well structured and supported by fairly up-to-date bibliographic references. I agree with the authors' statement "Factors that related to workplace violence more or less listed by some scholars which can be categorized into three classifications". However, recent studies show that there are many factors to consider. For example, in a recent literature review in the context of the emergency department, at least 24 theoretical models or conceptual frameworks are proposed in the literature that explains the phenomenon of violence against healthcare personnel (Ramacciati N, Ceccagnoli A, Addey B, Lumini E, Rasero L. Violence towards emergency nurses: A narrative review of theories and frameworks. Int Emerg Nurs. 2018 Jul;39:2-12. doi: 10.1016/j.ienj.2017.08.004.). Factors related to violence can be classified into at least 4 domains. In addition to the three correctly proposed by the authors, we must not forget the perpetrators of violent acts (patients, family members, companions, visitors or the health-care personnel for type 3 violence, called worker-on-worker violence). An example of categorization on 4 domains is the one proposed by Ramacciati and colleagues (see: Ramacciati N, Ceccagnoli A, Addey B, Rasero L. Violence towards Emergency Nurses. The Italian National Survey 2016: A qualitative study. Int J Nurs Stud. 2018 May;81:21-29. doi: 10.1016/j.ijnurstu.2018.01.017.)

#2 Page 16, lines 4-16: The results of the study are well discussed in the Discussion section. As regards the correlation between gender and the risks of exposure to violence, the data emerging from the literature are conflicting. For this, a comparison with the recent study published by Zhu and colleagues in the gynaecological and obstetric setting could also be useful. (see: Zhu L, Li L, Lang J. Gender Differences in Workplace Violence Against Obstetrics and Gynecologists in China: A National Congress Questionnaire. PLoS One. 2018 Dec 10; 13 (12): e0208693 . doi: 10.1371 /journal.pone.0208693.).

#3 Page 17, lines 3-7: I suggest to move in Conclusion section this sentence: "This research revealed that the prevalence of workplace violence was high. Government stakeholders and hospitals may give attention to reduce the burden of the problem for better health care in obstetrics, gynecology, and neonatal health. This can be address by drafting WPV incident reporting procedures, assign security personnel, making legislations and polices to make the health care place safe and conducive places both for patient and clients."

4# The list of references is numerically adequate and fairly up-to-date, however, it could be useful to enrich it with some more recent bibliographic sources.

5# The References section is full of errors, it needs to be thoroughly checked and corrected:

Please, will you to write the correct Journal name for the reference # 3. Paola F, Cecilia A and ,Rosaria D. Workplace violence in different settings and among various health professionals in an Italian general hospital: a cross-sectional study. Dovepress. 23 september 2016:1-3

Please, will you to write the starting and ending page numbers for the following references:

5.Sisawo EJ, Ouédraogo S, Huang SL. Workplace violence against nurses in the Gambia: mixed methods design. BMC Health Services Research. 2017;17.

7. Samir.N, Moustafa E. and Abou.H. Nurses’ attitudes and reactions to workplace violence in obstetrics and gynecology departments in Cairo hospitals. Eastern Mediterranean Health Journal. 2017;18(3).

8. Yenealem DG, Woldegebriel MK, Olana AT, Mekonnen TH. Violence at work: determinants & prevalence among health care workers, northwest Ethiopia: an institutional based cross sectional study.

Ann Occup Environ Med. 2019;31:8.

9. Abate. A, Abebaw. D, Birhanu. A , Zerihun. A, and Assefa. D.Prevalence and Associated Factors of Violence against HospitalStaff at Amanuel Mental Specialized Hospital in Addis Ababa, Ethiopia. Hindawi Psychiatry Journal Volume 2019

10. Fute M, Mengesha ZB, Wakgari N, Tessema GA. High prevalence of workplace violence among nurses working at public health facilities in Southern Ethiopia. BMC Nursing. 2015;14.

12.Lei Shi DZ, Chenyu Zhou,1Libin Yang, Tao Sun, Tianjun Hao, Xiangwen Peng, Lei Gao,1Wenhui Liu, Yi Mu, Yuzhen Han, Lihua Fan. A cross–sectional study on the prevalence and associated risk

factors for workplace violence against Chinese nurses. BMJ Global Health. 2017.

13. Shi L, Zhang D, Zhou C, Yang L, Sun T, Hao T, et al. A cross–sectional study on the prevalence and associated risk factors for workplace violence against Chinese nurses. BMJ Open. 2017;7(6).

16. Zhou C, Mou H, Xu W, Li Z, Liu X, Shi L, et al. Study on factors inducing workplace violence in Chinese hospitals based on the broken window theory: a cross-sectional study. BMJ Open. 2017;7(7).

17.Shafran-Tikva S, Zelker R, Stern Z, Chinitz D. Workplace violence in a tertiary care Israeli hospital - a systematic analysis of the types of violence, the perpetrators and hospital departments. Israel Journal

of Health Policy Research. 2017;6.

21. Bayram B, Çetin M, Çolak Oray N, Can İ. Workplace violence against physicians in Turkey’s emergency departments: a cross-sectional survey. BMJ Open. 2017;7(6).

22. Tiruneh BT, Bifftu BB, Tumebo AA, Kelkay MM, Anlay DZ, Dachew BA. Prevalence of workplace violence in Northwest Ethiopia: a multivariate analysis. BMC Nursing. 2016;15.

23. Abdellah RF, Salama KM. Prevalence and risk factors of workplace violence against health care workers in emergency department in Ismailia, Egypt. The Pan African Medical Journal. 2017;26.

24. Likassa.T,Gudissa.T, H/Mariam.Wand Jira. C. Assessment of Factors Associated with WorkplaceViolenceagainst Nurses among Referral Hospitals of Oromia Regional State, EthiopiaJournal of Health, Medicine and Nursing Vol.35, 2017.

26. Boafo IM. The effects of workplace respect and violence on nurses’ job satisfaction in Ghana: a cross-sectional survey. Human Resources for Health. 2018;16.

28. Cheung T, Lee PH, Yip PSF. Workplace Violence toward Physicians and Nurses: Prevalence and

Correlates in Macau. International Journal of Environmental Research and Public Health. 2017; 14(8).

This reference must be spelt out: 32. Amhara Regional State health bureau 2019

I hope these suggestions can help authors improve their article.

Reviewer #2: The authors present a cross-sectional study regarding an interesting topic. The investigation was conducted on a large sample (n=520) involving different professions, and the results have practical relevance. However, while their manuscript has some points of strength, there are several aspects that require attention before the paper can be considered for publication.

1) Throughout the manuscript there are many typos and unclear sentences; a linguistic revision is required. Some examples:

- Introduction, line 1: "define" should be "defined"; line 20, "burn out" should be "burnout"; last line: "is" should be "includes".

- Methods, line 16: "YES and NO" should be "YES or NO". Line 20: "number of staff" should be "number of staff members". Lines 23 and 27: "questioner" should be "questionnaire". Penultimate line: “a total of 08 persons”. Is this number correct, or should it be “8” or anything else?

- Results, section 3.2: "of them" should be deleted, "is" should be "was". Penultimate line: please define "OPD". Section 3.5: "sex" should be "gender" (the same applies to tables 1 and 3).

- Discussion, line 5: please rewrite "that was 20%". On the second page of the Discussion, from line 17 on: the sentences beginning with "This could be methodological similarity..." and "kinds of literature..." are not clear to me. Also, please don't say "a couple of articles" but specify their number.

- Table 3: I suggest to provide the data about monthly income in USD other than ETB, for clarity to an international audience.

2) The style of the references is inconsistent and some look incomplete: please refer to the authors guidelines of the Journal. In particular, number 3 presents names instead of surnames.

3) Statistical analysis is overall appropriate, but the authors should provide details about conditional odds ratios: since their main association index is AOR, what are the variables taken into account to adjust the raw OR?

4) A major concern regards adaptation of the questionnaire used for data collection: how exactly were the original tools modified, and why? Has any validation analysis been carried out? (e.g. Calculation of Cronbach’s alpha, exploratory factor analysis or PCA,…). As an alternative, are these data available elsewhere? The authors state that the Amharic version of the questionnaire was back-translated to English for the sake of consistency, but what information did the authors exactly obtain from this translation?

5) In table 3, please specify the years of training and differences between “diploma” and “degree”: do these educational level lead to different roles in the healthcare setting? I think that an international audience would find this information useful for comparing the results of this study to others.

6) In the discussion the authors cite papers coming from different countries, but only refer to healthcare providers as a whole. I think it should be useful to compare the findings of this study with data regarding specific professions, since a distinction between nurses, physicians and midwives has been done by the authors. Also, I suggest that if the authors wish to compare so many countries, they make sure that the role of the different professions involved is the same across all studies (e.g. in some countries midwives only attend uncomplicated vaginal delivery, and in others there are no nurses in the OBGYN units).

6. PLOS authors have the option to publish the peer review history of their article (what does this mean?). If published, this will include your full peer review and any attached files.

Reviewer #1: No

Reviewer #2: No

---

## [Author Response · Author response to Decision Letter 0]

27 Apr 2021

We really thank you. The comments were constractive and possible changes were made

---

## [Decision Letter · Decision Letter 1]

17 May 2021

PONE-D-20-30094R1

Working in labor and delivery unit increases the odds of work place violence in Amhara region referral hospitals: Cross-sectional study

PLOS ONE

Dear Dr. Dagnaw,

Thank you for submitting your manuscript to PLOS ONE. After careful consideration, we feel that it has merit but does not fully meet PLOS ONE’s publication criteria as it currently stands. Therefore, we invite you to submit a revised version of the manuscript that addresses the points raised during the review process.

We look forward to receiving your revised manuscript.

Kind regards,

Nicola Ramacciati

Academic Editor

PLOS ONE

Journal Requirements:

Additional Editor Comments (if provided):

Dear Authors,

I have just received reviewers' comments on your revised article.

Your commitment to following many of the suggestions proposed is evident.

There are only a few points to improve before your interesting work is published.

Here are the suggestions of the reviewers.

Editor:

#1 I suggest removing the title in each chart because it is described by its caption.

#2 The reference list should be checked for some errors in the citation style.

Reviewers' comments:

Reviewer's Responses to Questions

2. Is the manuscript technically sound, and do the data support the conclusions?

Reviewer #1: Yes

3. Has the statistical analysis been performed appropriately and rigorously? 

Reviewer #1: Yes

4. Have the authors made all data underlying the findings in their manuscript fully available?

Reviewer #1: Yes

5. Is the manuscript presented in an intelligible fashion and written in standard English?

Reviewer #1: Yes

6. Review Comments to the Author

Reviewer #1: Thanks for the opportunity to review this article. I am not a native speaker and I am not able to evaluate the grammatical and syntactic correctness of the manuscript. The text is clear and legible.

The authors followed many of the reviewers' suggestions. However there are some points that need to be corrected:

#1 Page 10: in sub-section "3.2. Organizational and working unit information" only two organizational aspects are presented: the Presence of policy for WPV and the Presence WPV dealing team. Explain that Table 2 also shows other data regarding the OBGYN departments organization and give a brief presentation of this data.

#2 Page 13: delete the title in the figure "Subjects of attackers".

#3 Page 3: the caption in figure 2 is incorrect. I suggest "Type of attackers in OBGYN departments of Amhara regional state referral hospitals."

#4 Page 16, lines 3-4: I do not completely agree with the authors on this point, already highlighted in the previous review. The authors' response to the Reviewer #1's suggestion was incorrect because type III violence does not affect retired workers, but all workers, who exercise these forms of violence on their co-workers due to mobbing, bullying, lateral hostility, negative interactions. This article talks about both type II violence (as rightly stated) but also type III violence (the so-called worker on worker violence).

#5 Page 17,I also suggest replacing the reference supporting this classification by using the main source of this classification which is the following and not the interesting and authoritative article by Phillips JP:

- University of Iowa Injury Prevention Research Center (UIIPRC). Workplace violence – A report to the nation. Iowa City, IA: University of Iowa; 2001

#6 Page 23: The references section still contains some errors in the citation style (many page numbers are missing, some author names are not indicated or abbreviated in the correct form). According to the editorial rules of PLOS One a DOI number for the full-text article is acceptable as an alternative to or in addition to traditional volume and page numbers but I suggest you check your references list by following these examples of format indicated by the editorial rules available at the following link: https://journals.plos.org/plosone/s/submission-guidelines#loc-manuscript-organization

- Hou WR, Hou YL, Wu GF, Song Y, Su XL, Sun B, et al. cDNA, genomic sequence cloning and overexpression of ribosomal protein gene L9 (rpL9) of the giant panda (Ailuropoda melanoleuca). Genet Mol Res. 2011;10: 1576-1588.

- Devaraju P, Gulati R, Antony PT, Mithun CB, Negi VS. Susceptibility to SLE in South Indian Tamils may be influenced by genetic selection pressure on TLR2 and TLR9 genes. Mol Immunol. 2014 Nov 22. pii: S0161-5890(14)00313-7. doi: 10.1016/j.molimm.2014.11.005.

I hope the authors find these tips useful in order to improve their interesting work.

7. PLOS authors have the option to publish the peer review history of their article (what does this mean?). If published, this will include your full peer review and any attached files.

Reviewer #1: No

---

## [Author Response · Author response to Decision Letter 1]

16 Jun 2021

Comment for editor: suggested points and comments are modified and corrected after all we authors were discussed deeply. Other ways we are thankful as obvious for working wit Us

Comment for academic reviewer: We are glad to receive your constructive comment and valuable comment and your comments are briefly discussed in response to reviewer part of the submission. We thank you

---

## [Editor Report · Decision Letter 2]

8 Jul 2021

Working in labor and delivery unit increases the odds of work place violence in Amhara region referral hospitals: Cross-sectional study

PONE-D-20-30094R2

Dear Dr. Dagnaw,

We’re pleased to inform you that your manuscript has been judged scientifically suitable for publication and will be formally accepted for publication once it meets all outstanding technical requirements.

Kind regards,

Nicola Ramacciati

Guest Editor

PLOS ONE
---

## [Editor Report · Acceptance letter]

7 Sep 2021

PONE-D-20-30094R2 

Working in labor and delivery unit increases the odds of work place violence in Amhara region referral hospitals: Cross-sectional study 

Dear Dr. Dagnaw:

I'm pleased to inform you that your manuscript has been deemed suitable for publication in PLOS ONE. Congratulations! Your manuscript is now with our production department. 

Kind regards, 

on behalf of

Dr. Nicola Ramacciati 

Guest Editor

PLOS ONE